# Absorption rate of subcutaneously infused fluid in ill multimorbid older patients

**Mathias Brix Danielsen**[1,2]*, **Lars Jødal**[3], **Johannes Riis**[1], **Jesper Scott Karmisholt**[2,4], **Óskar Valdórsson**[1], **Martin Gronbech Jørgensen**[1,2], **Stig Andersen**[1,2]

1 Department of Geriatric Medicine, Aalborg University Hospital, Aalborg, Denmark, 2 Department of Clinical Medicine, Aalborg University Hospital, Aalborg, Denmark, 3 Department of Nuclear Medicine, Aalborg University Hospital, Aalborg, Denmark, 4 Department of Endocrinology, Aalborg University Hospital, Aalborg, Denmark

* maad@rn.dk

## Abstract

### Background

Subcutaneous (SC) hydration is a valuable method for treating dehydration in the very old patients. Data are absent on the absorption rate, and the availability of SC infused fluid in the circulation in this group of patients where SC hydration is particularly relevant.

### Methods

We performed an explorative study on ill very old (range 78–84 years old) geriatric patients with comorbidities who received an SC infusion of 235 ml isotonic saline containing a technetium-99m pertechnetate tracer. The activity over the infusion site was measured using a gamma detector to assess the absorption rate from the SC space. The activity was measured initially every 5 minutes, with intervals extended gradually to 15 minutes. Activity in blood samples and the thyroid gland was measured to determine the rate of availability in the circulation.

### Results

Six patients were included. The mean age was 81 years (SD 2.1), the number of comorbidities was 4.6 (SD 1.3), and the Tilburg frailty indicator was 3.8 (SD 2.4). When the infusion was completed after 60 minutes, 53% (95% CI 50–56%) of the infused fluid was absorbed from the SC space, with 88% (95% CI 86–90%) absorbed one hour later. The absorption rate from the SC space right after the completion of the infusion was 127 ml/h (95% CI 90–164 ml/h). The appearance of the fluid into the blood and the thyroid gland verified the transfer from SC to circulation.

### Conclusion

This first explorative study of absorption of SC infused fluid in the very old found an acceptable amount of fluid absorbed from the SC space into the circulation one hour after infusion had ended. Results are uniform but should be interpreted cautiously due to the low sample size.

**Data Availability Statement:** All relevant data are within the paper and its Supporting information files. Data is anonymized.

**Funding:** The authors received no specific funding for this work.

**Competing interests:** The authors have declared that no competing interests exist.

## Trial registration

ClinicalTrials.gov Identifier: NCT04536324.

## Introduction

Dehydration is a herald of death [1, 2], and adequate fluid therapy is an important aspect of treating the older adult. Subcutaneous (SC) hydration is a safe and easy-to-apply method for parenteral fluid therapy recommended to treat mild dehydration and patients at risk of dehydration [3–5]. Previous studies have examined the absorption of SC hydration using radioisotopes to track fluid movement in younger adults [6] and healthy adults over 65 years [7, 8]. The studies found that absorption of the infused fluid was almost complete 60 minutes after the end of the infusion. However, SC hydration therapy is rarely relevant in healthy individuals, while it is useful in our older patients with concurrent comorbidities, poor physiological reserve, and few routes of pharmacological administration.

With SC hydration, the fluid is absorbed from the SC space into the capillaries through passive diffusion [9]. However, it has been shown that there is an increased leak from the capillaries during acute illness, potentially reducing their ability to absorb SC infused fluid [10]. Furthermore, albumin is the main osmotic component pulling the fluid into the capillaries [11]. Albumin is often reduced in the ill geriatric patient where SC hydration is relevant because of acute illness or malnutrition. Both of these physiological changes occur with advanced age and acute illness. However, the influence of these changes on the absorption rate and how complete the absorption is remains unknown for the ill geriatric patient with multimorbidity.

This led us to perform an explorative study on the ill, very old patients admitted to the hospital to estimate the time from infusion to availability in the circulation displayed as a fraction of the infused fluid found in the circulation in a clinically relevant population. We aimed to elucidate when SC hydration could be relevant and potentially guide clinicians in planning the older patient's hydration treatment.

## Methods

The study was approved by the local Committee on Health Research Ethics (Project ID: N–20200010) and was registered on Clinicaltrials.gov (NCT04536324). The study was conducted at Aalborg University Hospital, Aalborg, Denmark. The study was initially planned as a case-control study where the primary outcome was the difference between the absorption rate of ill versus non-ill older adults. We only completed the study on ill patients due to time limitations, restrictions from the COVID-19 pandemic, and our included patients' frailty. This paper reports all the secondary outcomes planned as registered on Clinicaltrials.gov.

### Participants

We recruited patients admitted to the local geriatric ward as a convenience sample. The study was designed to ensure the recruitment of a population where SC hydration is appropriate to support the study's clinical relevance and external validity [12].

Inclusion criteria were age above 75 years and ability to give informed consent. The capacity to provide informed consent was evaluated by the patient's physician and the study physician. This is in accordance with the ethical approval, which unfortunately excludes the

delirious patient, in which SC hydration might be especially suitable [3]. Exclusion criteria were: fluid restriction, risk of acute deterioration of illness, and very short life expectancy.

We collected data on the characteristics of the included patients from hospital charts (age, sex, number of prescriptions, number of comorbidities, Charlson Comorbidity Index (CCI) [13]) and through patient interviews (Tilburg Frailty Indicator [14]). Biochemical baseline characteristics recorded were: C-reactive protein, hemoglobin, sodium, potassium, urea nitrogen, creatinine, osmolality, albumin, and eGFR (CKD-EPI [15]). These were obtained by routine analysis at the hospital laboratory on the day of the study procedures.

## Study setup

We used technetium-99m ($^{99m}$Tc) pertechnetate as a marker for the movement of the infused fluid from the SC space to the circulation as its uptake from SC tissue has been documented to mimic SC water uptake [8].

We gave the SC infusion through a butterfly needle (BD Saf-T-Intima™—22G, Becton, Dickinson, and Company, Franklin Lakes, New Jersey, USA) inserted on the left side of the abdomen, and we collected blood samples through an indwelling intravenous catheter (BD Venflon™ Pro Safety—18G, Becton, Dickinson, and Company, Franklin Lakes, New Jersey, USA) inserted into the antecubital vein. Patients were infused with 235 ml of isotonic saline (Sodium chloride 0.9%, B. Braun, Melsungen, Germany). 30 MBq of $^{99m}$Tc were mixed into the infusion fluid before starting the infusion. Mixing pertechnetate into the fluid from the start, rather than using bolus injection(s) at a specific time point(s), ensures that the measured activity is representative of the fluid distribution, even if the uptake rate should be different in the early and late part of the infusion.

After baseline activity measurements were recorded at the insertion site at time 0, we started the SC infusion. The initial infusion speed was 125 ml/h, and the infusion rate was increased to 250 ml/h after 10 minutes if the patients did not experience discomfort. An infusion pump managed the infusion rate (CODAN 717V, CODAN ARGUS AG, Baar, Switzerland).

The infusion was completed in 1 hour. The high infusion rate was chosen to reduce the duration of the study to ensure the included ill patients could complete the observation period. During the study, the activity over the infusion site was measured at 5, 10, 15, 20, 30, 40, 50, 60, 70, 85, 100, 115, 130, and 145 minutes after the start of infusion by us using a gamma detector (Captus® 3000, Capintec, 7 Vreeland Road, Florham Park, New Jersey, USA). At the same time points, blood samples of 2.7 ml were taken to measure the activity in the circulation. Before extraction of each of these blood samples, 2.7 ml of blood was taken as waste blood [16]. After extracting each blood sample, the catheter was rinsed with 5 ml of isotonic saline. Also, pertechnetate activity measurements were performed over the thyroid gland, as pertechnetate is rapidly absorbed by the thyroid gland [17, 18]. These measurements were taken from 20 minutes after the start of the infusion and with a similar interval as those taken over the infusion site.

## Method of measurement

The total dose infused $^{99m}$Tc was measured by a dose calibrator (CRC-15R®, Capintec, 7 Vreeland Road, Florham Park, New Jersey, USA) before being mixed with the infusion fluid. The activity was measured both over the infusion site and the thyroid gland at a distance of 30 cm using a gamma detector. All activity measurements with the gamma detector were done with a counting time of 30.0 seconds. The blood samples taken during the study were analyzed by a dedicated gamma counter (2480 Wizard²™ Gamma Counter, PerkinElmer, Waltham, Massachusetts, USA). All activity measurements were decay corrected to the start of the infusion.

## Sample size

The sample size was based on a previous study on healthy older adults [7]. They reported an absorption constant of 2.29 hour$^{-1}$ with a standard deviation of 0.3. We speculated a difference in the absorption constant between the ill and non-ill on 15% (0.345 hour$^{-1}$). With an alpha of 0.05 and a power of 80% (beta = 0.2), we would need six patients according to our calculations (one-sided, two-sample paired means t-test, STATA 16). As noted at the beginning of this Methods section, the circumstances with the inclusion of multimorbid, ill patients did not allow the case-control part of the study to be performed. However, we saw a justification for the study, and we had to balance the discomfort and strain on these ill patients against the number of participants needed. We thus kept the sample size as calculated without additional numbers as we would get sufficient data from 6 patients, based on the previous study on healthy older adults [8].

## Statistical analysis and calculations

Categorical variables are presented using numbers and percentages, and continuous variables are presented as mean and standard deviation (SD) as the data are without outliers.

For each patient, we calculated a conversion factor at the infusion site to convert measured activity (counts per second, cps) to ml of infused fluid. This patient-specific conversion factor was calculated using the slope of the initial linear part of the activity curve (0–10 minutes). We use the 0–10 minutes value to reduce error from the amount of fluid already absorbed. We calculated the fraction of the infused fluid still present in the subcutaneous space at time $t$ after the end of the infusion:

$$\text{volume}_{SC} = \text{measured activity} \times \text{patient-specific conversion factor}$$

$$\text{fraction present in the SC space} = 100\% \times (\text{volume}_{SC}/235 \text{ ml})$$

We estimated the absorption rate in ml/min specifically for our infusion rate in the 10 minutes following the infusion's completion from the reduction in activity over time at the infusion site:

$$\text{absorption rate} = (\text{volume}_{SC} \text{ at } 70 \text{ min} - \text{volume}_{SC} \text{ at } 60 \text{ min})/10 \text{ min}.$$

As the absorption rate in ml/min varies over the study (dependent on the amount of fluid infused but not yet drained), many studies report the absorption constant (k) instead. The theoretical relation between the absorption rate and the absorption constant is:

$$\text{absorption rate (ml/min)} = \text{absorption constant (min}^{-1}) \times \text{present volume}_{SC} \text{ (ml)}$$

where absorption constant in min$^{-1}$ can be turned into hour$^{-1}$ by multiplying by 60, e.g. k = 0.02 min$^{-1}$ = 1.2 hour$^{-1}$. Absorption rate estimates are presented as means. To validate that the absorption rate measured over the SC space did represent a transfer to the bloodstream with availability to body physiology, we also measured the uptake into the blood and the thyroid gland. We calculated the mean time to 50% absorbed (half-time, t$_{\frac{1}{2}}$) using exponential regression on log-transformed data, corresponding to expecting an exponential decay. To allow for deviations from a purely exponential form, fitting with a quadratic term was also performed. For thyroid and blood data, a sigmoid curve form (probit function, inverse normal) was used to describe the overall shape. The regression analysis on measurements from the SC space was done from the 60-minute mark and onwards. For data from blood and thyroid, it is done from the start of infusion. We report the mean value for the half-time with 95% confidence interval derived from the half-time for the individual participants.

To estimate the potential effect of albumin on the absorption the log absorption rate after 60 minutes was analyzed in random intercept by participant regression. The regression was adjusted by gender, time, and quadratic time, and P-values were found at the regression effects.

The collected data were stored using REDCap version 7.0.11 hosted at Aalborg University Hospital [19].

All analyses were done using STATA 16 (StataCorp. 2019. Stata Statistical Software: Release 16. College Station, TX: StataCorp LLC.) and Microsoft Excel 365® Microsoft 2020©.

## Results

We recruited six patients, three men and three women, from September to November 2020. See Fig 1 for the flowchart. The mean age of the patients was 81 years (SD 2.1), the mean CCI score was 1.8 (SD 1.3), and the mean number of prescription drugs before admission was 10 (SD 4.1). Two patients had a Tilburg frailty indicator over five (judged as frail) [14, 20]. All baseline measurements can be found in Table 1.

None of the patients experienced any adverse reaction during or after the infusion, and all infusions were completed after 60 minutes. In one patient, the indwelling catheter for collecting blood samples clotted after 40 minutes, and further blood samples could not be drawn.

As expected, the infusion site's activity measurements showed that fluid accumulated in the SC space during the infusion (0–60 minutes). At the end of infusion, the mean volume of fluid still present in the SC space was 111 ml (SD 7.8) of the 235 ml of infused, corresponding to 53% (95% CI 50–56%) of the infused fluid having been absorbed. The fraction absorbed after 25, 40, 55, 70, and 85 minutes after the completion of the infusion was 74% (95% CI 70–77%), 81% (78–84%), 85% (82–88%), 88% (86–90%), and 90% (87–92%), respectively (Fig 2).

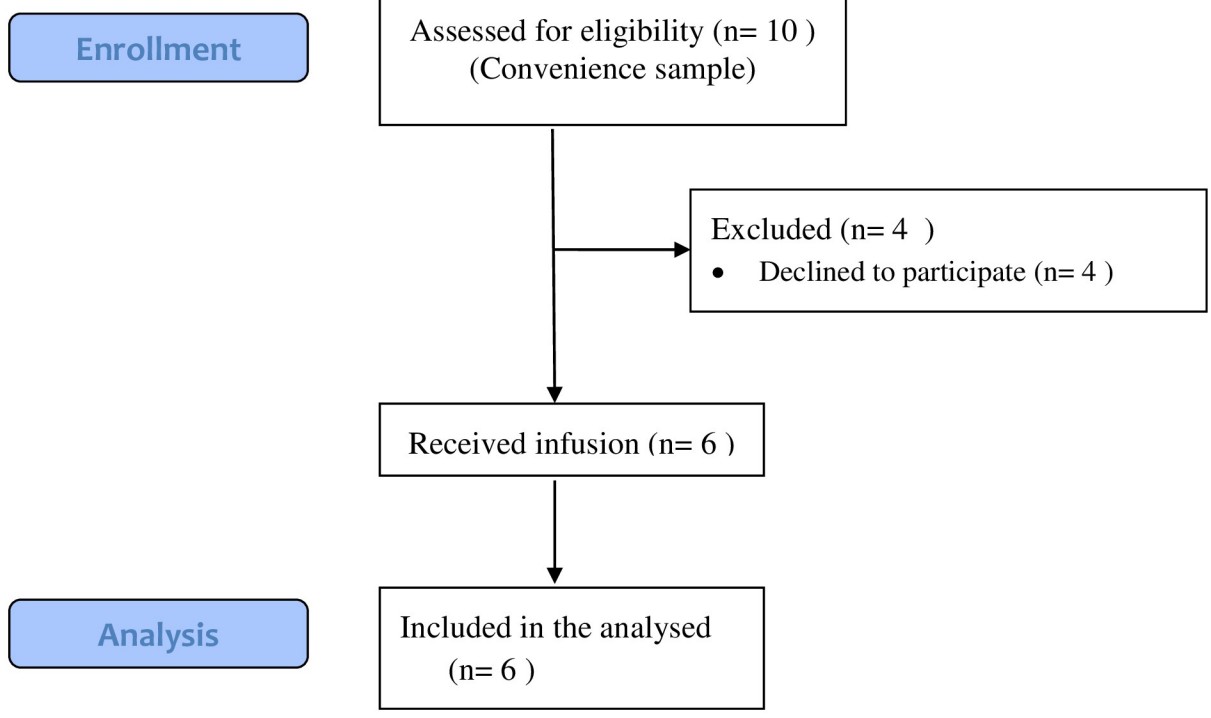

**Fig 1. Flowchart of participants.**

**Table 1. Baseline values of the six patients.**

| | Mean (SD) |
|---|---|
| **Number of patients** | 6 |
| **Age** | 81 (2.1) |
| **Sex, female (No/ percent)** | 3/50% |
| **Number of known comorbidities** | 4.6 (1.2) |
| **Charlson Comorbidity Index** [13] | 1.8 (1.3) |
| **Tilburg frailty indicator** [14] | 3.8 (2.4) |
| **Number of prescription drugs** | 10 (4.1) |
| **Treated with anti-coagulant medication** | 1 (16.7%) |
| **Systolic Blood Pressure (mm Hg)** | 122 (9.8) |
| **Diastolic Blood Pressure (mm Hg)** | 71 (5.7) |
| **Pulse (/min)** | 81 (21) |
| **C-reactive protein (mg/l)** | 62 (38) |
| **Hemoglobin (mmol/l)** | 6.2 (0.6) |
| **Sodium (mmol/l)** | 141 (1.6) |
| **Potassium (mmol/l)** | 3.9 (0.2) |
| **Urea (mmol/l)** | 9.3 (2.2) |
| **Creatinine (µmol/l)** | 97 (42) |
| **eGFR (ml/min/1.73m$^2$)** | 62 (24) |
| **Albumin (g/l)** | 28 (4.2) |
| **Osmolality (mmol/kg)** | 297 (5.5) |

Graphical representation of the mean percentage of infused fluid absorbed over time across all six patients. The X-axis is in minutes after the start of the infusion. The infusion was complete after 60 minutes. The vertical brackets represent a 95% confidence interval. The Y-axis is in percentage.

Figs 3–5 shows the activities at the infusion site, in blood and uptake in the thyroid gland, respectively. We calculated absorption rates in ml/minute to present an easily interpretable

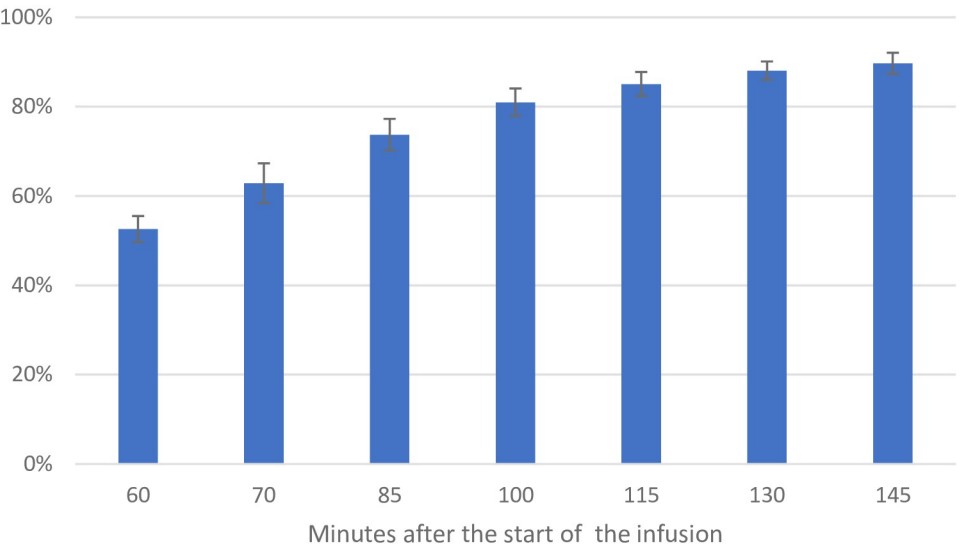

**Fig 2. Mean percentage of fluid absorbed over time across the six patients.**

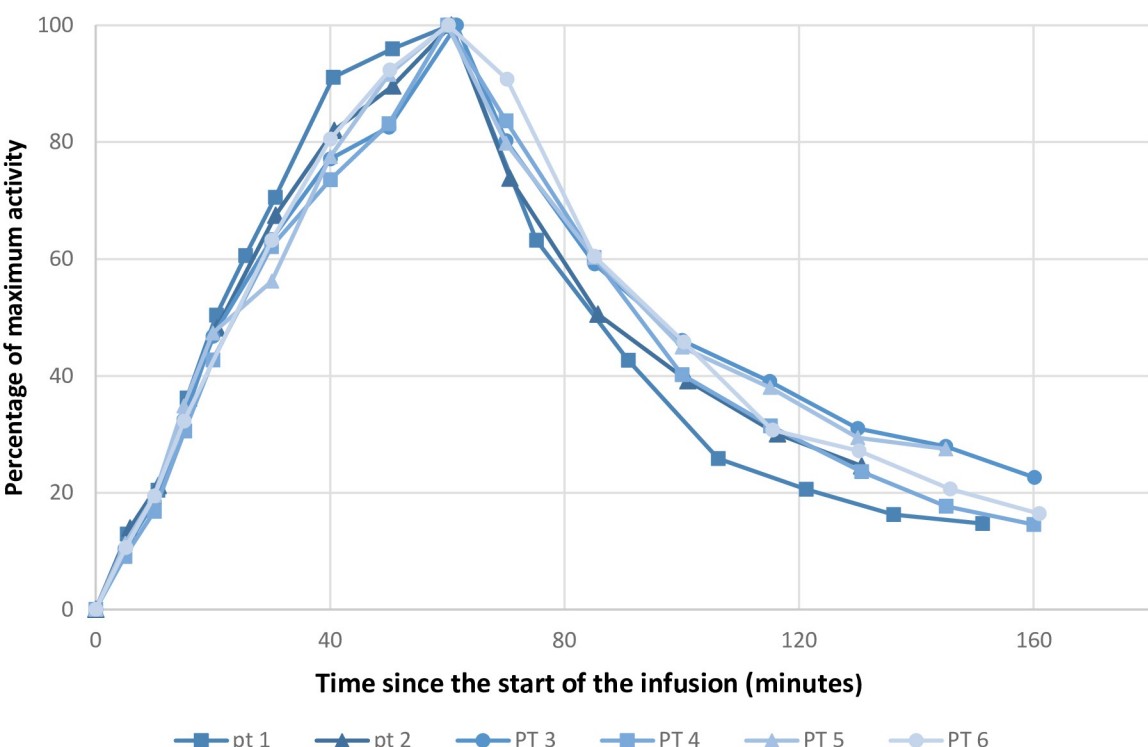

**Fig 3. Activity at the infusion site over time.** Abbreviation: PT: Patient. Graphical representation of the activity over the infusion site. The infusions ended after 60 minutes. All data points are normalized to a percentage of the maximum value of a given series. The X-axis is in minutes after the start of the infusion. The Y-axis is in percentage of maximum activity.

absorption rate. These numbers are specific for our setup (235 ml infused over 60 minutes) but provide an example of achievable absorption rates. The mean absorption rate estimated from measurement at the infusion site was 127 ml/h (95% CI 90–164 ml/h).

Exponential regression (regression on the logarithm of the values) without a quadratic term found a mean value of the absorption constant $k = 1.12$ (SD = 0.12) hour$^{-1}$. This corresponds to a half-time $t_{1/2} = 0.693/k = 0.62$ hour = 37 minutes (95% CI 34–42 minutes). Including the quadratic term and calculating $t_{1/2}$ as the time where the original value had dropped to 50% found $t_{1/2} = 31$ minutes (27–35 minutes, S1 Fig in S1 File), i.e., shorter but of similar magnitude.

The blood data showed that 50% of the plateau value was reached after 48 minutes (43–52 minutes, S2 Fig in S1 File), and in the thyroid gland data, it was reached after 58 minutes (56–60 minutes, S3 Fig in S1 File).

Statistical analysis of absorption rate versus serum level of albumin found a statistically significant regression effect ($p = 0.02$), with increasing absorption rate with increasing albumin levels. However, this effect's size cannot be calculated with a meaningful result due to the low number of included patients.

## Discussion

We conducted an explorative study to describe the absorption rate and availability in the circulation for fluid given through an SC catheter in the ill, very old (range 78–84 years old), multimorbid, hospitalized geriatric patients. To our knowledge, this is the first study to explore this

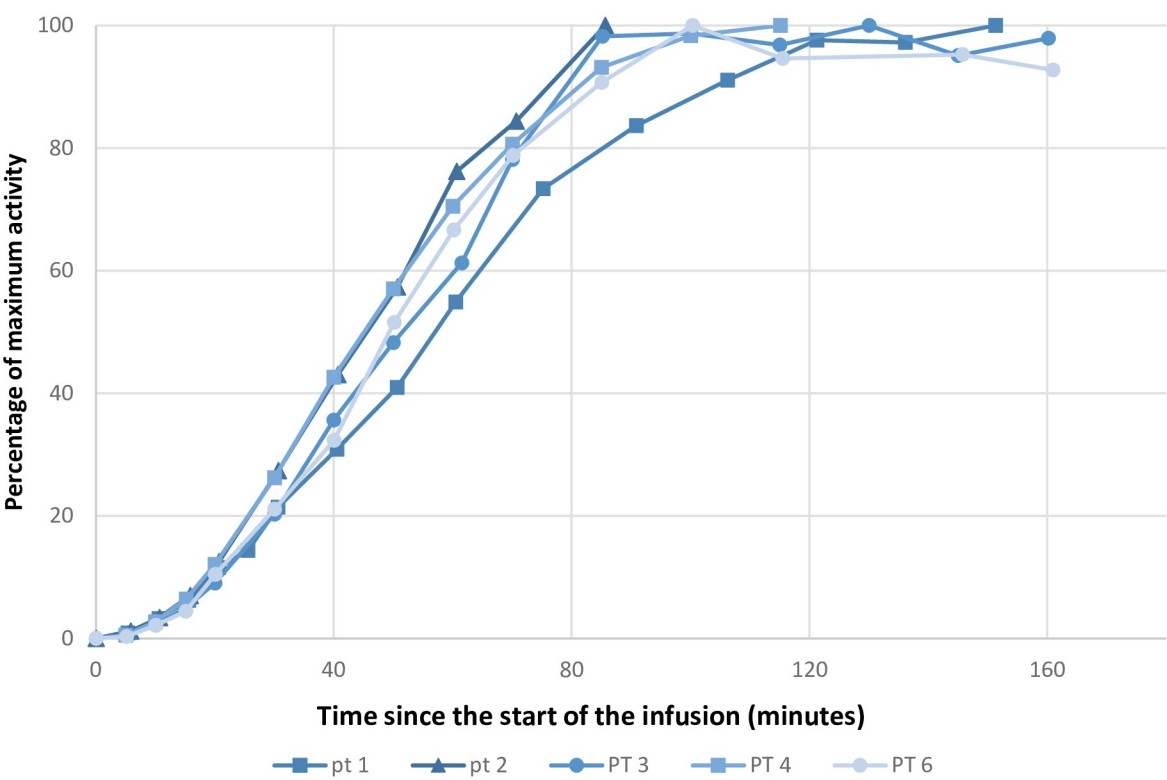

**Fig 4. Activity in the blood over time.** Abbreviation: PT: Patient. Graphical representation of the activity in the blood. The infusions ended after 60 minutes. All data points are normalized to a percentage of the maximum value of a given series. The X-axis is in minutes after the start of the infusion. The Y-axis is in percentage of maximum activity. Data from patient number 5 is missing as the indwelling catheter for the collection of blood samples clotted.

type of hydration in this vulnerable patient group. With an infusion of 235 ml over 60 minutes, we found an average absorption rate from the SC space of 127 ml/hour (95% CI 90–164 ml/h) right after the end of the infusion. The rate, however, will depend on the individual setup (e.g. infusion rate, fluid type), but our setup demonstrated that an absorption rate of around 127 ml/hour is achievable in illgeriatric patients. Furthermore, our measurements on the blood samples and over the thyroid gland confirm that the infused fluid does enter the bloodstream, rather than just spreading locally within the SC tissue.

In more general terms, regardless of infusion rate, our data indicate that half of the fluid remaining in the SC space after completion of an infusion will be absorbed in about 31 minutes. This number increases slightly to about 37 minutes if a purely exponential function is assumed (absorption constant $k = 1.12$ hour$^{-1}$). Such absorption half-lives are markedly longer than the previous study on healthy adults over 65 years that report a half-life of only 18 minutes [7].

The absorption constant k can indicate the maximum fluid accumulation in the SC space based on the infusion rate. Assuming a purely exponential function, the volume of fluid accumulating at the infusion site will slowly approach a maximum volume equal to the infusion rate divided by k. This allows us to calculate the expected fluid accumulation in a clinical setting with a slower infusion rate than used in our study.

The standard recommendation on SC fluid infusion describes that 1 liter of fluid can be administered subcutaneously over 8–10 hours [12, 21]. Assuming 8 hours, this corresponds to

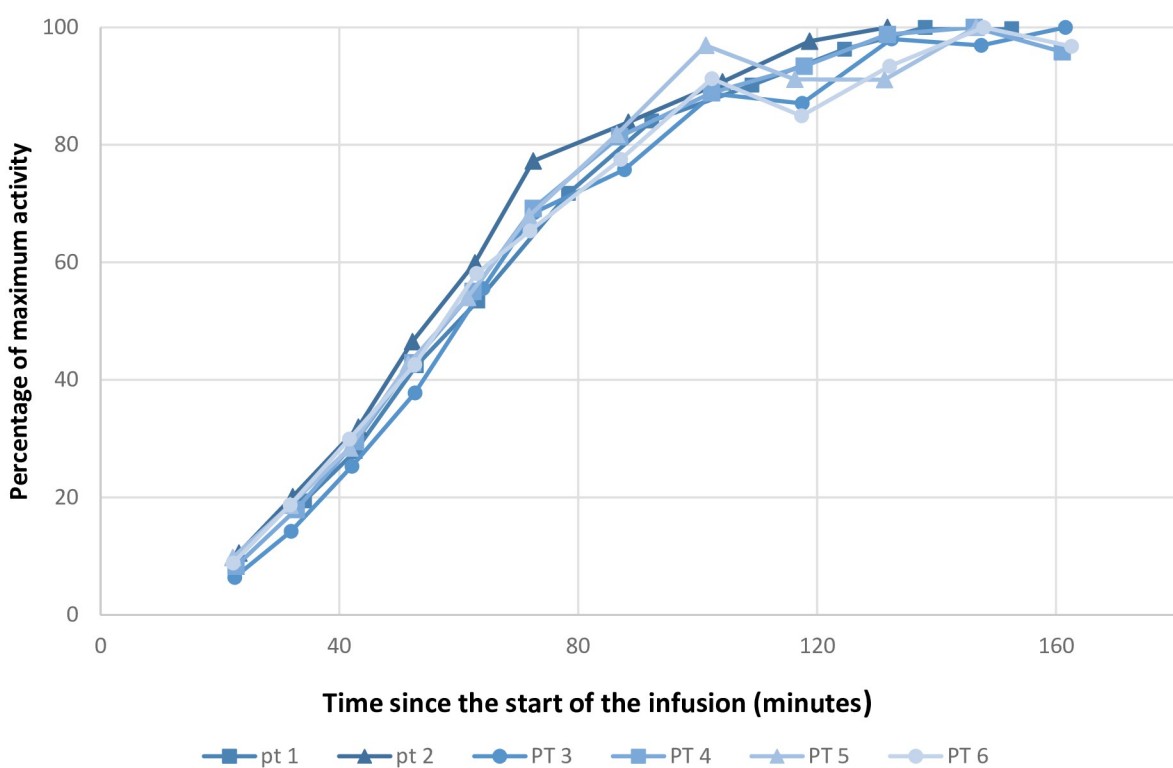

**Fig 5. Activity in the thyroid gland measured over time.** Abbreviation: PT: Patient. Graphical representation of the activity over the thyroid gland. The infusions ended after 60 minutes. All data points are normalized to a percentage of the maximum value of a given series. The X-axis is in minutes after the start of the infusion. The Y-axis is in percentage of maximum activity.

125 ml/hour. A value of $k = 1.12$ hour$^{-1}$ then theoretically predicts a maximum volume of 125 ml/hour / 1.12 hour$^{-1}$ = 111 ml being temporarily accumulated in the SC space, regardless of how long this infusion continues. Of course, extrapolation outside the studied time interval should always be handled with care. Accordingly, the important point here is not the specific volume of 111 ml but that the accumulated volume is small enough to corroborate this recommendation as a safe procedure in clinical practice, also for ill, older adults.

With 85% (95% CI 86–90%) of infused fluid absorbed 55 minutes after the end of infusion, the completeness of absorption is lower than reported in studies on a young population aged 21–35. Here, 95% of infusion fluid was absorbed after 45 minutes after the end of the infusion [6], and in healthy older adults aged over 65, no residual activity was found 60 minutes after the end of infusion. The latter study, however, used hyaluronidase that aided the absorption of SC hydration [8]. Our study found that there was still around 10% of the infused fluid retained in the SC space one and a half-hour after the infusion's completion. Our data did not extend beyond this time point as the procedures wore out our ill patients. All patients requested to be transferred back to the ward at this point.

The important finding of 90% absorbed leaves 10% retained fluid, which is less clinically relevant when treating the mildly dehydrated patients as SC hydration prescriptions are often made in round numbers [3].

We found a statistically significant effect of albumin on the absorption rate, with increased albumin levels increasing the absorption rate. This finding is as expected, but further studies with more participants are required to estimate the size of this effect.

Our study showed that activity measurements over the thyroid gland could be used as a qualitative confirmation that the infused fluid has become part of body physiology without requiring intravenous cannulation.

To sum up, our findings showed that in our group of ill, very old patients (range 78–84 years old) geriatric patients with multiple comorbidities, SC hydration worked well, even though—as expected—the absorption of the fluid was slower than found in younger populations. The radiotracer measurements showed that the infused liquid gradually left the infusion site. The blood samples and measurements over the thyroid gland confirmed that the fluid had indeed reached the blood and become systemic. Theoretical calculations on the expected local volume accumulation (125 ml/h for an indefinite time) in the SC space resulted in reassuringly low numbers; however, this needs to be confirmed in future studies.

## Limitations

As the radioactive tracer is distributed in the body, the activity measurement will include a background signal from already-distributed activity. However, the detector's collimation ensures that it measures only locally, i.e., a small fraction of the whole body. For this reason, the background signal will be only a tiny fraction of the measured signal and be without notable influence on our results. Our study infused isotonic saline, and the absorption rate may differ with other fluid types. The amount of fluid infused is lower than typically used in a clinical setting. The absorption rate and residual fluid in the SC space could be different if 500 ml or 1000 ml were infused over a longer duration. We had planned to have the patients return for a second procedure eight weeks after discharge to investigate the difference between acutely ill and not acutely ill. Due to time limitations, restrictions from the COVID-19 pandemic, and patient frailty, this was not feasible. This paper reports the most important outcomes, namely those on the acutely ill older adults. These outcomes were all listed in our clinicaltrial.gov registration. Finally, our sample was relatively small, with just six patients, but results were marked and uniform in all patients, conforming to a reliable absorption portrayal.

In conclusion, we found clinically useful absorption rates from the SC space of around 127 ml/hour right after the end of the infusion in geriatric inpatients using our infusion setup. The absorption was slower compared to previous studies in younger patients. However, only a small proportion of the infusion fluid remained in the subcutaneous space one hour after completion of the infusion. To guide clinicians, our results corroborate the clinical experience that one liter of fluid can be administered and absorbed satisfactorily in the very old, ill, multimorbid patient. Our results are uniform, but the limited sample size encourages further studies to corroborate our results.

## Supporting information

**S1 File.**
(PDF)

**S1 Protocol. Study protocol in Danish.**
(PDF)

**S2 Protocol. Study Protocol translated to English.**
(PDF)

**S1 Dataset. Anonymised data set.**
(XLSX)

**S1 Checklist. CONSORT 2010 checklist of information to include when reporting a randomised trial.**
(DOCX)

## Acknowledgments

Statistical assistance by Niels Henrik Bruun, MSc, Unit of Clinical Biostatistics, Aalborg University Hospital, regarding curve fitting is thankfully acknowledged.

## Author Contributions

**Conceptualization:** Mathias Brix Danielsen, Lars Jødal, Jesper Scott Karmisholt, Martin Gronbech Jørgensen.

**Data curation:** Mathias Brix Danielsen, Lars Jødal.

**Formal analysis:** Mathias Brix Danielsen, Lars Jødal.

**Investigation:** Mathias Brix Danielsen, Johannes Riis, Óskar Valdórsson.

**Methodology:** Mathias Brix Danielsen, Lars Jødal, Johannes Riis, Jesper Scott Karmisholt, Martin Gronbech Jørgensen, Stig Andersen.

**Project administration:** Mathias Brix Danielsen.

**Resources:** Lars Jødal, Jesper Scott Karmisholt, Stig Andersen.

**Supervision:** Jesper Scott Karmisholt, Stig Andersen.

**Visualization:** Mathias Brix Danielsen, Lars Jødal.

**Writing – original draft:** Mathias Brix Danielsen, Lars Jødal, Johannes Riis, Stig Andersen.

**Writing – review & editing:** Mathias Brix Danielsen, Lars Jødal, Johannes Riis, Jesper Scott Karmisholt, Óskar Valdórsson, Martin Gronbech Jørgensen, Stig Andersen.

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
