## [Decision Letter · Decision Letter 0]

12 Jul 2022

PONE-D-21-38016Absorption Rate of Subcutaneously Infused Fluid in Ill Multimorbid Older PatientsPLOS ONE

Dear Dr. Danielsen,

Thank you for submitting your interesting manuscript to PLOS ONE. The comments of reviewers are below. After careful consideration, therefore, we feel that it has merit but does not fully meet PLOS ONE’s publication criteria as it currently stands. We invite you to submit a revised version of the manuscript that addresses the points raised during the review process.

We look forward to receiving your revised manuscript.

Kind regards,

Antony Bayer

Academic Editor

PLOS ONE

Journal Requirements:

2. Please describe in your methods section how capacity to provide consent was determined for the participants in this study. Please also state whether your ethics committee or IRB approved this consent procedure. If you did not assess capacity to consent please briefly outline why this was not necessary in this case.”.

Reviewers' comments:

Reviewer's Responses to Questions

**Comments to the Author**

1. Is the manuscript technically sound, and do the data support the conclusions?

Reviewer #1: Yes

Reviewer #2: Yes

Reviewer #3: Yes

2. Has the statistical analysis been performed appropriately and rigorously? 

Reviewer #1: Yes

Reviewer #2: Yes

Reviewer #3: Yes

3. Have the authors made all data underlying the findings in their manuscript fully available?

Reviewer #1: Yes

Reviewer #2: No

Reviewer #3: Yes

4. Is the manuscript presented in an intelligible fashion and written in standard English?

Reviewer #1: Yes

Reviewer #2: Yes

Reviewer #3: Yes

5. Review Comments to the Author

Reviewer #1: nice article and interesting one, i have the following points:

Introduction:

written well and objectives were clear

Methods:written well and detailed

1) why did only include 6 patients knowing that you will not include any control as per the initial study design? was it based on any calculations or just an arbitrary number? no details were mentioned on sample size part on how it was calculated

2) did use any pump for the infusion or only gravity?

3) based on what did use that infusion rate 250 ml per hour?? as in clinical practice that's very quick and i doubt anyone will use it in clinical setting unless in emergency situation

Results:

written well and detailed

Discussion:

written well and detailed

i think you should add how your study results can help physicians on treating similar patients other than what we already knows

Reviewer #2: A small exploratory research study (n=6) was conducted which aimed to describe the absorption rate of subcutaneous infused fluid in a select group of patients. Upon completion of the infusion (at 60 minutes), 53% of the infused fluid was absorbed; an hour later it was 88%. The absorption rate was 127 ml/h immediately after the completion of the infusion.

Minor revisions:

1- Abstract: Provide 95% confidence intervals for the 53%, 88% and the rate of 127 ml/h.

2- Line 151: A beta of 0.2 corresponds to a power of 80%. Perhaps the beta is misspecified. Indicate the statistical testing method which achieves the specified alpha and power.

3- Considering that the sample size is small and the distribution of the data from small sample sizes cannot be shown to be normal, it is standard practice to summarize these types of data using median, first and third quartiles.

4- Table 1: Specify the percent male.

5- Line 263: Specify the statistical testing method used to estimate the p-value for the significant correlation. Perhaps a graph might illustrate the correlation better than specifying the p-value or providing an effect size.

Reviewer #3: Although it is a small study, the sample size is sufficient to demonstrate the ability of the subcutaneous tissue to absorb liquids,

therefore, it is an interesting and useful study that is a little closer to the reality that many of us see in daily clinical practice: elderly patients with concurrent comorbidities, poor physiological reserve, and few routes of pharmacological administration.

The introduction encompasses the problem efficiently, the methodology and results are correct and well developed.

The main limitation is the administered volume, which is scarce, and it only reflects the initial absorption capacity of the subcutaneous tissue.

As fluid is infused through this route, accumulation is progressive and the residual volume itself limits absorption capacity, reducing absorption speed and tolerance. Although it is described in the limitations, it is very important because it does not allow us to extrapolate these results to clinical where infusion of larger volumes is often necessary.

Another limitation of the study is found in the type of patients included, since the limited variety of conditions makes it impossible to assess other aspects that may influence absorption apart from hypoalbuminemia or hypotension.

It would be interesting if they described a little more what solution they are referring to, when speaking of isotonic saline, it is intuited that it is the "normal" 0,9% saline, but any balanced crystalloid could respond to that description.

This could be relevant because there are differences in local tolerance between crystalloids and possibly altered absorption speed (although theoretically of little relevance, it is in these patients where every detail counts).

However, I consider a good, interesting, and practical study that allows us to take another step towards the reality of many patients, so I recommend it to be considered for publication

6. PLOS authors have the option to publish the peer review history of their article (what does this mean?). If published, this will include your full peer review and any attached files.

Reviewer #1: No

Reviewer #2: No

Reviewer #3: No

---

## [Author Response · Author response to Decision Letter 0]

19 Aug 2022

We have provided a point-by-point response to every comment by the referees in the document named Response to Reviewers. I was unable to change the order of the uploaded files. Therefore, our response to the reviewers are at the end of the pdf file.

---

## [Decision Letter · Decision Letter 1]

4 Sep 2022

PONE-D-21-38016R1Absorption rate of subcutaneously infused fluid in ill multimorbid older patientsPLOS ONE

Dear Dr. Danielsen,

Thank you for submitting your revised manuscript to PLOS ONE and your careful attention to the previous reviewer comments. After further consideration, we feel that it has considerable merit but does not fully meet PLOS ONE’s publication criteria as it currently stands. Therefore, we invite you to submit a revised version of the manuscript that addresses the points raised below. You will see that Reviewer 2 has now highlighted a misspelling of "percentage" and also asked for the method used for calculating confidence intervals. 

Please can you ensure also that you use consistent terminology to describe the patients you studied. For example, your title describes them as “ill multimorbid…”, then the background of your abstract refers to “geriatric patients”, the methods to “frail, ill octogenarians with comorbidities” and the conclusion to simply “octogenarians”. There is similar variation throughout the text. They do seem to have been ill and multimorbid and on a geriatric ward, but I am not sure about all being frail or octogenarians?

For example, at line 221 you state that a TFI over two was judged as frail. That two patients had a score of two or less would seem to suggest not all were “frail”? Can you also provide a reference for this TFI cut-off (or correct it)? I thought a TFI over five or six indicating frailty was usual?   

Similarly, were all the patients octogenarians (i.e., aged 80-89)? I note recruitment was of those over 75 and mean age 81 (sd2.1).

You refer variously to the Tilburg Frailty Indicator/Scale/Score. Please use “Indicator” and TFI-score consistently.

“Data” are plural – please correct as necessary e.g., line 210 and 294 (and maybe others).

Line 276 – …thyroid gland AT 58 minutes…

We look forward to receiving your revised manuscript.

Kind regards,

Antony Bayer

Academic Editor

PLOS ONE

Journal Requirements:

Reviewers' comments:

Reviewer's Responses to Questions

**Comments to the Author**

1. If the authors have adequately addressed your comments raised in a previous round of review and you feel that this manuscript is now acceptable for publication, you may indicate that here to bypass the “Comments to the Author” section, enter your conflict of interest statement in the “Confidential to Editor” section, and submit your "Accept" recommendation.

Reviewer #2: (No Response)

Reviewer #3: All comments have been addressed

2. Is the manuscript technically sound, and do the data support the conclusions?

Reviewer #2: Yes

Reviewer #3: Yes

3. Has the statistical analysis been performed appropriately and rigorously? 

Reviewer #2: Yes

Reviewer #3: Yes

4. Have the authors made all data underlying the findings in their manuscript fully available?

Reviewer #2: No

Reviewer #3: Yes

5. Is the manuscript presented in an intelligible fashion and written in standard English?

Reviewer #2: Yes

Reviewer #3: Yes

6. Review Comments to the Author

Reviewer #2: Minor revision

1- Graphs: Percentage is misspelled.

2- Table 1: Percent is misspelled.

3- State the statistical methods used to estimate the 95% confidence intervals.

Reviewer #3: (No Response)

7. PLOS authors have the option to publish the peer review history of their article (what does this mean?). If published, this will include your full peer review and any attached files.

Reviewer #2: No

Reviewer #3: No

---

## [Author Response · Author response to Decision Letter 1]

14 Sep 2022

We have provided a point-by-point response to every comment by the referees in the document named Response to Reviewers. I was unable to change the order of the uploaded files. Therefore, our response to the reviewers are at the end of the pdf file.

---

## [Editor Report · Decision Letter 2]

26 Sep 2022

Absorption rate of subcutaneously infused fluid in ill multimorbid older patients

PONE-D-21-38016R2

Dear Dr. Danielsen,

Thank you for your careful attention to revising the manuscript. We’re pleased to inform you that it has been judged scientifically suitable for publication and will be formally accepted for publication once it meets all outstanding technical requirements.

Kind regards,

Antony Bayer

Academic Editor

PLOS ONE
---

## [Editor Report · Acceptance letter]

29 Sep 2022

PONE-D-21-38016R2 

Absorption rate of subcutaneously infused fluid in ill multimorbid older patients 

Dear Dr. Danielsen:

I'm pleased to inform you that your manuscript has been deemed suitable for publication in PLOS ONE. Congratulations! Your manuscript is now with our production department. 

Kind regards, 

on behalf of

Professor Antony Bayer 

Academic Editor

PLOS ONE